# Effects of Carbazole Derivatives on Neurite Outgrowth and Hydrogen Peroxide-Induced Cytotoxicity in Neuro2a Cells

**DOI:** 10.3390/molecules24071366

**Published:** 2019-04-07

**Authors:** Yoshiko Furukawa, Atsushi Sawamoto, Mizuki Yamaoka, Makiko Nakaya, Yuhzo Hieda, Tominari Choshi, Noriyuki Hatae, Satoshi Okuyama, Mitsunari Nakajima, Satoshi Hibino

**Affiliations:** 1Department of Pharmaceutical Pharmacology, College of Pharmaceutical Sciences, Matsuyama University, 4-2 Bunkyo-cho, Matsuyama, Ehime 790-8578, Japan; asawamot@g.matsuyama-u.ac.jp (A.S.); 16131209@matsuyama-u.ac.jp (M.Y.); mu.yakuri.015@gmail.com (M.N.); sokuyama@g.matsuyama-u.ac.jp (S.O.); mnakajim@g.matsuyama-u.ac.jp (M.N.); 2Faculty of Pharmacy and Pharmaceutical Sciences, Fukuyama University, 1 Sanzo, Gakuen-cho, Fukuyama, Hiroshima 729-0292, Japan; hieda@fukuyama-u.ac.jp (Y.H.); choshi@fukuyama-u.ac.jp (T.C.); satoshihibino@ma.ccnw.ne.jp (S.H.); 3School of Pharmaceutical Sciences, Health Science University of Hokkaido, Ishikari, Tobetsu, Hokkaido 061-0293, Japan; nhatae@hoku-iryo-u.ac.jp

**Keywords:** carbazole, neuro2a cells, neurite outgrowth, neuronal death, anti-oxidant

## Abstract

Many studies have demonstrated that oxidative stress plays an important role in several ailments including neurodegenerative diseases and cerebral ischemic injury. Previously we synthesized some carbazole compounds that have anti-oxidant ability in vitro. In this present study, we found that one of these 22 carbazole compounds, compound **13** (3-ethoxy-1-hydroxy-8- methoxy-2-methylcarbazole-5-carbaldehyde), had the ability to protect neuro2a cells from hydrogen peroxide-induced cell death. It is well known that neurite loss is one of the cardinal features of neuronal injury. Our present study revealed that compound **13** had the ability to induce neurite outgrowth through the PI3K/Akt signaling pathway in neuro2a cells. These findings suggest that compound **13** might exert a neurotrophic effect and thus be a useful therapy for the treatment of brain injury.

## 1. Introduction

Carbazole alkaloids, which are produced by plants such as those of the genus *Murraya* or *Clausena* [1], have recently attracted much attention because 1) they have various abilities such as anti-microbial, anti-tumor, anti-pileptic, anti-histaminic, anti-oxidant and anti-inflammatory actions [2]; and 2) they are being used as lead compounds for drug development [3,4,5]. Recently, we synthesized some carbazole derivatives having anti-oxidant ability, which was evaluated in terms of their radical-scavenging activity against 2,2-diphenyl-1-picrylhydrazyl (DPPH) and/or 2,2′-azinobis-(3-ethylbenzhiazolic-6-sulfonate) cations (ABTS) [5,6].

As the results of several recent studies have indicated that neurological disorders are linked to elevated levels of oxidative stress [7,8] and that anti-oxidant(s) might have therapeutic potential [9], we investigated herein, using neuroblastoma neuro2a cells, whether our carbazole compounds could exert an anti-oxidant effect on neuronal cells. Neuro2a cells are a mouse neural crest-derived cell line and are frequently used to study neuroprotective abilities of various factors [10,11]. As a generator of reactive oxygen species (ROS), we used hydrogen peroxide (H_2_O_2_), which is known to easily penetrate into cells and to generate high levels of ROS [12,13].

Neuro2a cells are also extensively used to study neuronal differentiation and neurite growth [14]. Neurite outgrowth is known to be crucial for neuronal plasticity and neuronal regeneration [15], and these actions are considered to be important for developing therapies to promote neuronal regeneration in the case of nerve injury and neurological disorders [16]. We thus evaluated the ability of our carbazole compounds to promote neurite outgrowth from neuro2a cells.

As a result, we successfully found that one of our carbazole derivatives had this ability as well as anti-oxidant ability; and so, we then investigated the regulatory mechanisms of the neurite outgrowth triggered by this compound. Regarding the regulatory mechanisms at play in neurite outgrowth from neuronal cells (including not only neuro2a cells but also rat PC12 cells), various signal pathways have been reported to be involved, such as extracellular signal-regulated kinase (ERK) [17,18,19], Akt and phosphatidylinositol 3-kinase (PI3K) [16,20]. Thus, we investigated whether our carbazole compound had the ability to directly activate (phosphorylate) any signal transduction molecule(s) and whether the blockage of such signal transduction(s) would reduce this carbazole compound-induced neurite outgrowth from neuro2a cells.

## 2. Results and Discussion

### 2.1. Effects of Carbazole Derivatives on H_2_O_2_-Induced Cell Death of Neuro2a Cells

First, we examined whether our carbazole derivatives (Figure 1) might exhibit a protective effect against H_2_O_2_-induced cell death of neuro2a cells cultured in normally used medium, namely, medium containing 10% fetal bovine serum (FBS). For this experiment, cells were seeded in wells of a 96-well plate and maintained in this medium for 24 h. The cells were then treated with test compounds (10 µM concentration of each compound) for 1 h and then incubated for an additional 18 h in the presence of H_2_O_2_ (30 µM). After the exposure to H_2_O_2_, the cell viability was significantly (**p* < 0.05) reduced to about 80 %, but some of the compounds (**3**, **5**, **13**, **21,** and **22**) had the protective effect on H_2_O_2_-induced cell viability reduction in this experimental condition and gave similar cell viability in comparison to that of control and 50 µM vitamin E (V.E)-treated cells (Figure 2).

### 2.2. Effects of Carbazole Derivatives on Neuronal Differentiation of Neuro2a Cells

To investigate whether our carbazole derivatives could induce neurite outgrowth from neuro2a cells, we first cultured the cells for 24 h in 24-well plates in a medium containing 10% serum. Then they were cultured in a low-serum (2% FBS) medium for 24 h to induce a transition from the proliferation phase to the differentiation phase, after which the cells were incubated for 48 h in a low-serum medium containing test samples (0.3–5 µM). As a positive control, *N*^6^,2′-O-dibutyryl- adenosine 3′,5′-cyclic monophosphate (db cAMP), a membrane-permeable cAMP analog, was used at the concentration of 5 mM.

Figure 3A shows the untreated cells (none) having a round shape with few neurites and the db cAMP-treated ones apparently displaying long neurites. As shown in Table 1, one of the compounds tested (**13**) at the wide range of concentrations apparently induced neurite extension from neuro2a cells after a 48-h treatment; and some of the compounds (compounds **4**, **5**, **6**, **8**, **9**, **10**, **12**, **18**, **19**, **20**, and **22**) at the limited range of concentrations slightly induced neurite extension after the 48 h treatment, whereas others (compounds **1**, **2**, **4** and **12**) were toxic during the incubation period. Figure 3A also shows the representative photographs of the cells cultured with 0.5 µM compound **13** for 48 h. Figure 3B shows that the % of neurite-bearing cells after the 48-h incubation with no compound, 5 mM db cAMP and 0.5 µM compound **13** was 56.8 ± 6.75 % (*** *p* < 0.001 vs. control [none]), and 23.1 ± 4.75 % (*** *p* < 0.001), respectively.

### 2.3. Effects of Carbazole 13 on H_2_O_2_-Induced Cell Death of Neuro2a Cells

As we revealed that compound **13** (3-ethoxy-1-hydroxy-8-methoxy-2-methylcarbazole-5- carbaldehyde) had the protective effect on H_2_O_2_-induced cell viability reduction (Figure 2) and neuritogenic (Figure 3) activities against neuro2a cells, we then examined in detail the protective effect of compound **13** against H_2_O_2_-induced apoptosis of neuro2a cells cultured in low-serum medium. For this experiment, cells in wells of a 96-well plate were cultured in the normal-serum medium for 24 h and then in the low-serum one for 24 h. After that the cells were treated with compound **13** at the concentration of 1–10 µM or vitamin E at the concentration of 10–50 µM for 1 h, and then incubated for an additional 18 h in the presence of H_2_O_2_ (30 µM). Figure 4 shows that the cells in the low-serum medium were more sensitive to oxidative stress than those in the normal-serum medium (Figure 2), namely, 30 µM H_2_O_2_ reduced the cell viability to 36.1 ± 14.1 % (*** *p* < 0.001) of the none-treated group after 18 h. On this condition, the anti-oxidative abilities of 1 or 5 µM compound **13** against 30 µM H_2_O_2_ were almost equal to those of 10 µM vitamin E, and that of 10 µM compound **13** against 30 µM H_2_O_2_ were almost equal to that of 50 µM vitamin E.

### 2.4. Signaling Pathway Involved in Compound 13-Induced Neurite Outgrowth from Neuro2a Cells

Next, we characterized the cell signaling pathway functioning in the compound **13**-induced neurite outgrowth. Over the past 2 decades, numerous studies have revealed the molecular mechanisms underlying the regulation of neurite outgrowth by various factors. The representative pathway for neurite outgrowth is the activation of the PI3K/Akt and/or protein kinase A (PKA)/mitogen-activated protein kinase (MAPK)/ERK signaling pathways [16,17,18,19,20,21]. We thus tested by immunoblot analysis whether compound **13** had the ability to promote the phosphorylation of ERK1/2 and/or Akt in neuro2a cells. For this experiment, cells seeded in 6-well plates were maintained in a normal-serum medium for 24 h and then in a low-serum medium for 24 h. Thereafter, the cells were kept for 0.5–5 h in a low-serum medium containing 30 µM compound **13**.

ERK1/2 is a component of the MAPK signaling cascade. This MAPK/ERK pathway is activated by a variety of extracellular agents, including growth factors, hormones, and also cellular stresses to trigger cellular processes that include mainly proliferation and differentiation; and it also contributes to the control of a large number of other cellular processes, including synaptic plasticity, e.g., long-term potentiation (LTP) in hippocampal neurons [22,23]. As ERK2 (42 kDa), but not ERK1 (44 kDa), has been suggested to be the one mainly involved in neurogenesis and cognitive function [22], we analyzed the ratio of phosphorylated ERK2 (pERK2) to total ERK2 (ERK2) in the present study. Figure 5A shows that treatment with compound **13** apparently induced the phosphorylation of ERK2 in neuro2a cells when the cells were incubated in the presence of compound **13** (30 µM) for 3 h or more.

Akt, a serine/threonine-specific protein kinase, is known to be protein kinase B (PKB). Akt is a key downstream effector of PI3K, and the PI3K/Akt pathway is a signal transduction pathway that protects neurons against various forms of injury in the brain [24]. Phosphorylated Akt inhibits downstream targets, such as proapoptotic protein and caspases [25]. Figure 5B shows that treatment with compound **13** resulted in time-dependent phosphorylation of Akt in neuro2a cells when the cells were treated with compound **13** (30 µM).

These results revealed that compound **13** had the ability to induce the phosphorylation of both ERK1/2 and Akt. In order to confirm whether compound **13** induced neurite outgrowth through MAPK/ERK and/or PI3K/Akt signaling pathways, we preincubated the cells for 30 min with specific inhibitors of ERK1/2 (10 µM U0126), PI3K (10 µM LY294002) or PKA (1 µM H89) and then incubated them with compound **13** for 48 h. These inhibitors themselves had no effect on the formation of neurites (data not shown). Figure 6 shows that the blockade of PI3K by LY294002 and the blockade of PKA by H89 reduced the neurite-promoting effect of compound **13**. On the other hand, the blockade of MAPK/ERK by U0126 scarcely affected the formation of neurites. These results suggest that compound **13**-induced neurite outgrowth from neuro2a cells was being regulated through the PI3K/Akt-mediated signaling pathway and not the PKA/MAPK/ERK one. Other various signal pathways such as c-jun N-terminal kinase (JNK) have been reported to be involved in neurite outgrowth of neuro2a cells [19,21]. We will investigate whether not only PI3K/Akt-mediated signaling pathway, but also other signaling pathway(s) underlie the compound **13**-induced neuronal differentiation or not.

Many recent studies revealed that oxidative stress directly and/or indirectly results in neuronal death [9] and neurodegeneration in the brain [26], and also plays a crucial role in various neurodegenerative diseases such as Alzheimer’s disease (AD), Parkinson’s disease (PD), multiple sclerosis, and amyotrophic lateral sclerosis (ALS) [7,8]. In fact, edaravone, a free-radical scavenger, is a clinical drug for the treatment of ischemic stroke [9].

On the other hand, recent studies also showed that factors regulating neurite outgrowth might be targets for treating aging-induced or lesion-induced neurological dysfunctions, because aging/lesion is associated with neuron atrophy and impaired sprouting [27]. Thus, attention has been recently drawn to compounds promoting neurite outgrowth as being candidates of neuroregenerative drugs [28]. In fact, compounds promoting neurite outgrowth reportedly enhance learning and memory of aged subjects [21,29,30] or exhibit a potent antidepressant effect [31].

Taken together our findings suggest that compound **13** (3-ethoxy-1-hydroxy-8-methoxy- 2-methylcarbazole-5-carbaldehyde), by having both anti-oxidant ability together and neurogenetic ability, might be beneficial for the treatment of neurological disorders. In fact, various series of carbazole derivatives have been designed and synthesized as neuroprotective therapeutic agents for AD [32], PD [33], ALS [34], ischemic stroke [35] or traumatic brain injury [36] and so on in recent years. We are planning to investigate whether compound **13** has neuroprotective ability in vivo using various disease model mice.

## 3. Materials and Methods

### 3.1. Chemical and Reagents

Carbazole derivatives were provided as follow; Compounds **2** (2-hydoxycaebazole: No. 322-55321) and **4** (4-hydroxycarbazole: No. H1014) were purchased from FUJIFILM Wako Pure Chemical Corp. (Tokyo, Japan) and Tokyo Chemical Ind. Co., Ltd., (Tokyo, Japan), respectively. Compounds **1**, **3**, **5**, **6** (carazostatin), **20**, **21**, and **22** were synthesized by the method previously reported [6,37]. Compound **7** was synthesized by the method previously reported [38]. Compounds **8**, **9**, **10**, **11**, and **12** ((±)carquinostatin A) were synthesized by the method previously reported [3]. Compounds **13**, **14**, **15**, **16**, **17**, **18** (carbazomadurin A), and **19** ((*S*)-(+)-carbazomadurin B) were synthesized as reported earlier [4].

All compounds were dissolved in dimethyl sulfoxide (DMSO) to yield 30 mM stock solutions. LY294002 from Wako Pure Chemical Ind., Ltd. (Osaka, Japan), U0126 and H-89 from Calbiochem Corp. (San Diego, CA USA), and vitamin E from Combi-Blocks Inc. (San Diego, CA, USA) were also dissolved in DMSO. Db cAMP sodium salt was purchased from Sigma-Aldrich Company Ltd. (St. Louis, MO, USA) and dissolved in phosphate-buffered saline (PBS).

### 3.2. Cell Culture

Neuro2a cells were maintained and cultured as previously described [39]. All cell culture materials such as Dulbecco’s Modified Eagle Medium (DMEM), FBS, and antibiotics were purchased from Thermo Fisher Scientific (Waltham, MA, USA). The final concentration of DMSO in all culture media was below 0.1%.

### 3.3. Determination of Cell Viability

For the experiment described in Section 2.1, neuro2a cells were seeded in wells of a 96-well plate at a density of 1 × 10^4^ cells/well; whereas for the experiment described in Section 2.3, the cell density was 2 × 10^3^ cells/well. Cell viability was determined by the MTT assay as previously described [40].

### 3.4. Assessment of Neurite Outgrowth

Neuro2a cells were seeded in wells of a 24-well plate at a low density (5 × 10^3^ cells/well). Neurites were defined as cellular processes with lengths equivalent to 1 or more diameters of the cell body. The percentage of cells bearing neurites was calculated as the % of number of neurites divided by the total number of cells examined. More than 100 cells were randomly chosen for counting neurite-bearing cells.

### 3.5. Immunoblot Analysis

Neuro2a cells were seeded in wells of a 6-well plate at a density of 2.5 × 10^5^ cells/well. The cell extracts were prepared as previously described [39]. Equal amounts of proteins (20 μg) were separated on SDS-polyacrylamide gels and electroblotted onto an Immuno-Blot^TM^ PVDF membrane (Bio-Rad Laboratories, Hercules, CA, USA). As primary antibodies, rabbit polyclonal antibodies against 44/42 ERK1/2, which recognize 44-kDa ERK1 and 42-kDa ERK2; phospho-44/42 MAPK (Thr202/Tyr204), specific for phosphorylated ERK1/2 (pERK1/2); Akt; and phosphor-Akt (Ser473), which recognize phosphorylated Akt (pAkt), were purchased from Cell Signaling Technology Inc. (Woburn, MA, USA). As secondary antibody, alkaline phosphatase-linked anti-rabbit IgG (Cell Signaling) was used. Immunoreactive bands were detected by use of the NCB/BCIP reagent (Roche Diagnotics GmbH, Mannheim, Germany).

### 3.6. Statistical Analysis

All results were expressed as means ± SEM. Significant differences of experiments with two groups were analyzed by the Student’s *t*-test. Experiments with three or more groups were subjected to a one-way ANOVA followed by the Dunnet’s multiple comparison test. *p* < 0.05 was taken to be statistically significant.

## 4. Conclusions

In conclusion, our present study revealed using neru2a cells that carbazole **13** (3-ethoxy-1-hydroxy-8-methoxy-2-methylcarbazole-5-carbaldehyde) had the ability to protect neuro2a cells against H_2_O_2_-induced death and also to induce neurite outgrowth from these cells by acting through, at least, the PI3K/Akt signaling pathway.

## Figures and Tables

**Figure 1 molecules-24-01366-f001:**
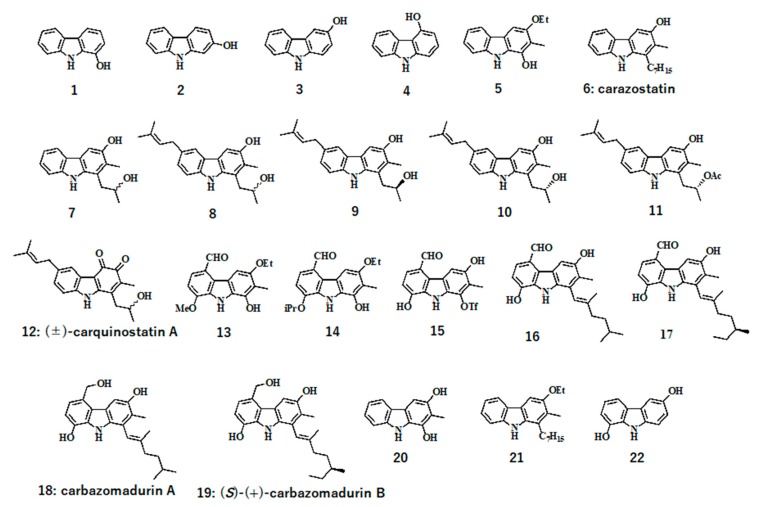
Structures of carbazole derivatives.

**Figure 2 molecules-24-01366-f002:**
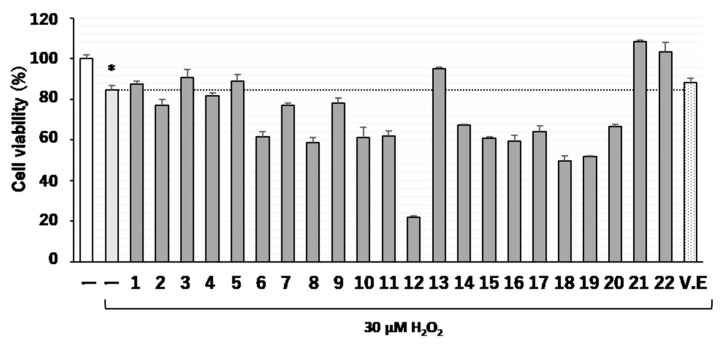
Effects of carbazole derivatives on neuro2a cell viability. Neuro2a cells pretreated with test compounds (compounds **1**–**22** at the concentration of 10 µM or with vitamin E [V.E] at the concentration of 50 µM) were exposed to 30 µM H_2_O_2_. The results represent the mean ± SEM (*n* = 5, different culture). Significance difference in values between the none-treated and H_2_O_2_-treated cells: * *p* < 0.05 (Student’s *t* test).

**Figure 3 molecules-24-01366-f003:**
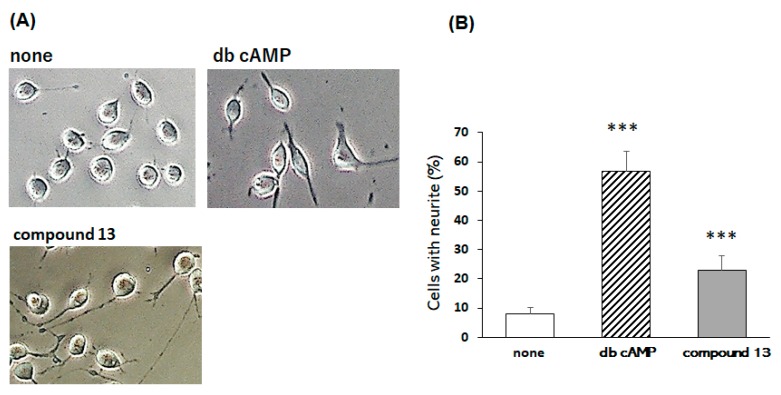
Effects of compounds **13** and dibutyryl cAMP (db cAMP) on neurite outgrowth from neuro2a cells. Neuro2a cells were treated with test compounds (5 mM db cAMP or 0.5 µM compound **13**) for 48 h, and then morphological images were captured by phase-contrast microscopy (**A**). Cells were randomly chosen for counting neurite-bearing cells (*n* = more than 100 cells per group) (**B**). Significance difference in values between the sample-treated and non-treated cells: *** *p* < 0.001 (one-way ANOVA followed by the Dunnett’s multiple comparison test).

**Figure 4 molecules-24-01366-f004:**
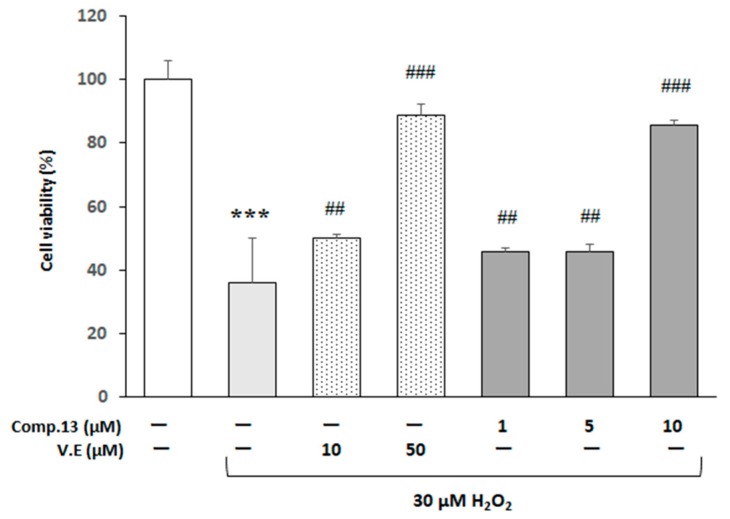
Effects of compound **13** and vitamin E on neuro2a cell viability. Neuro2a cells pretreated with test compounds (compound **13** at the concentration of 1–10 µM or vitamin E at the concentration of 10–50 µM) were exposed to 30 µM H_2_O_2_. The results represent the mean ± SEM (*n* = 5, different culture). Symbols are significantly different for the following condition; vs. none-treated cells (*** *p* < 0.001, Student’s *t* test), vs. 30 µM H_2_O_2_-treated cells (^##^
*p* < 0.01, ^###^
*p* < 0.001, one-way ANOVA followed by the Dunnett’s multiple comparison test).

**Figure 5 molecules-24-01366-f005:**
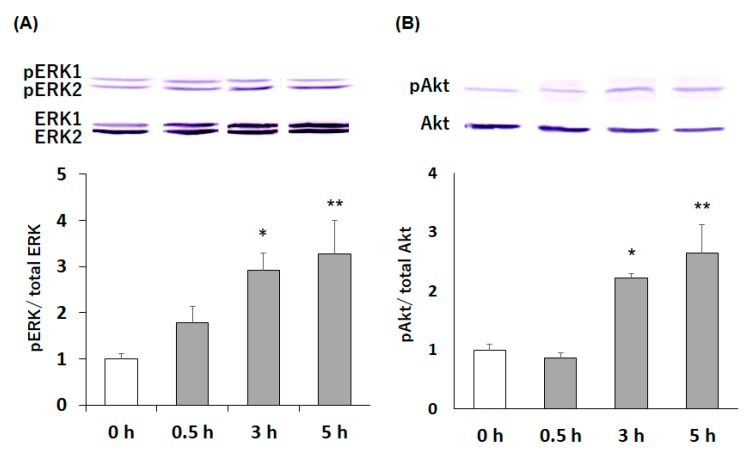
Effects of compound **13** on activation of extracellular signal-regulated kinases (ERK) 1/2 (**A**) and Akt (**B**) in neuro2a cells. Neuro2a cells were treated with 30 µM compound **13** for 0.5, 3 or 5 h; and then equal amounts of protein were analyzed by immunoblot analysis. The results represent the mean ± SEM (*n* = 3, different culture). Significance difference in values between compound **13**-treated and non-treated (0 h) cells: * *p* < 0.05, ** *p* < 0.01 (one-way ANOVA followed by the Dunnett’s multiple comparison test).

**Figure 6 molecules-24-01366-f006:**
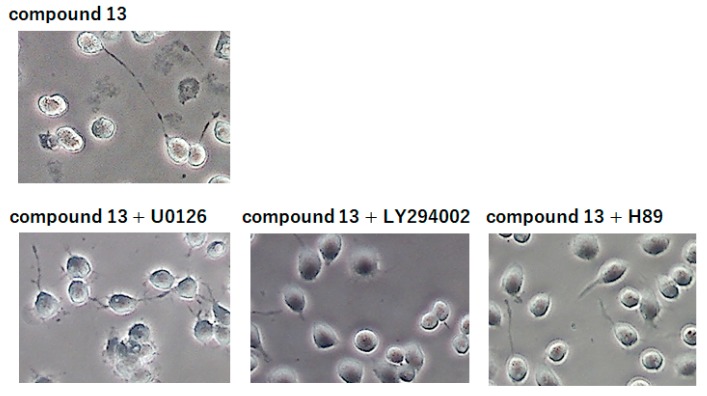
Effects of compound **13** on activation of Akt in neuro2a cells. Neuro2a cells were treated with each inhibitor for 30 min and then incubated with 0.5 µM compound **13** for 48 h.

**Table 1 molecules-24-01366-t001:** Effects of carbazole derivatives on neurite outgrowth in neuro2a cells. Cells were treated with carbazole derivatives at the indicated concentrations for 48 h.

Compounds	0.3 μM	0.5 μM	1.0 μM	3.0 μM	5.0 μM
**1**	−			cell death	
**2**	−			cell death	
**3**	−			−	
**4**	±			cell death	
**5**	±			−	
**6**	±			±	
**7**	−			−	
**8**	±	±	+	+	±
**9**	±			−	
**10**	±			−	
**11**	−			−	
**12**		+	+		cell death
**13**	+	+	+	+	+
**14**	−			−	
**15**	−			−	
**16**	−			−	
**17**	−			−	
**18**	±			−	
**19**			±		
**20**	±			±	
**21**	−			−	
**22**	−			+	

(−) No cells with neurite outgrowth; (±) cells with slight neurite outgrowth; (+) cells with apparent neurite outgrowth; (cell death) most of the cells died.

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
