# Peer review of "Effects of Carbazole Derivatives on Neurite Outgrowth and Hydrogen Peroxide-Induced Cytotoxicity in Neuro2a Cells"

_molecules, 2019, doi:10.3390/molecules24071366_

Round 1

Reviewer 1 Report

The present manuscript “Effects of carbazole derivatives on neurite outgrowth and hydrogen peroxide-induced cytotoxicity in neuro2a cells”, by Furukawa Y et. al have shown the ability of compound 13(3-ethoxy-1-hydroxy-8- 24 methoxy-2-methylcarbazole-5-carbaldehyde), in protecting neuro2a cells from 25 hydrogen peroxide-induced cell death and mediating neuronal differentiation. In the present manuscript the authors have claimed that compound 13 (3-ethoxy-1-hydroxy-8- 24 methoxy-2-methylcarbazole-5-carbaldehyde) and other carbazole derivatives protect cell death and mediate cell differentiation. In addition, they claim that, this protective effect is mediated via PI3/ ERK1/2 kinase pathways.

The manuscript is clearly and concisely written; however, I have major concerns and comments and is not suitable for publication in the present form. They are;

·        The authors have reported that the carbazole derivatives 13, 21 and 22 have neuroprotective effects. However, the data looks different. There are many issues in the Figure -2. First the 30mM H2O2 alone induced only 16% cell death to start with in comparison to control (open bar), which was inhibited by the compound 13 (~12%). Compound 21 (8%) and compound 22 (4%) mediated neuroprotective effect in comparison to control cells (open bar). Why the authors continued this experiment to evaluate the carbazole derivatives since H2O2 induced only 16% cell death?

·        In my opinion, compound 12 was more potent among all derivatives in the presence of H2O2 to inhibit cell viability. Compounds 6, 8, 10, 11, and 14 - 20 were equally potent in inhibiting cell viability.

·        In Figure 3, the micrographs didn’t represent the bar graph shown on the right side.

·        In Figure 4, H2O2 decreased cell viability more than 50%, which is different than the data shown in Figure 2. Why the authors have used higher doses of H2O2 when it already reached maximum cell viability inhibition. I am not sure the authors have performed dose response of H2O2 induced inhibition of cell viability?

·        In Figure 5 the western blot data is not the representative blot coming from a single gel but rather from more that one gel. Now a days very good pERK and pAKT antibodies are available to probe with for excellent results.

·        Finally, the authors didn’t discuss how the compound 13 is mediating its protective effect via PI3/ ERK kinases. Whether this compound 13 is binding to any G-protein coupled receptors, growth factor receptors or directly activating PI3/ ERK kinase pathways?

Author Response

The answer to Reviewer 1:   

According to the kind suggestion of reviewer 1, we collected our manuscript as follows.

1) About “the 30 mM H2O2 alone induced only 16 % cell death…  Why the authors continued this experiment to evaluate the carbazole derivatives since H2O2 induced only 16% cell death?”;

For the experiment for Figure 2, neuro2a cells cultured in the presence of 10% FCS were exposed to 30 μM H2O2. In this condition, cells were in the growing state, and the cell’s sensitivity to oxidative stress was much lower (cell viability was reduced by 30 μM H2O2 to 80 %). When neuro2a cells were cultured in the presence of 2% FCS (Figure 4), cells stopped growing and were more sensitive to oxidative stress (cell viability was reduced by 30 μM H2O2 to 36 %).

We employed the growing state for the experiment of Figure 2 in order to study the anti-oxidative effect of carbazole compounds in the less damage situation of the cells. In order to show that 30 μM H2O2 exerted significantly oxidative stress on neuro2a cells, we added the p value (*p<0.05) in the figure and text (p.2 lines 69-70). Thanks to your pointed out.

2) About “In my opinion, compound 12 was more potent among all derivatives in the presence of H2O2 to inhibit cell viability.”;

As pointed by reviewer, compound 12 gave the most potent influence on the cell viability (most highly toxic). But we tried to explore the carbazole compounds with anti-oxidative effect.

We omitted the p value for the sample (compounds 13, 21, and 22)-treated cells vs. H2O2-treated cells in figure 2, and replaced with the sentence “some of the compounds (3, 5, 13, 21, and 22) exerted a protective effect against H2O2-induced death of the cells equal to or greater than that of 50 µM vitamin E (V.E), a positive control.” (p.2 lines 70-72).

3) About “In Figure 3, the micrographs didn’t represent the bar graph shown on the right side.”;

As we showed the representative micrographs in (A). We took pictures of the field of view of the microscope where a lot of cells with neurite as far as possible were present. Numeric values of the cells with neurite was shown in (B). As a result, the micrographs (A) didn’t represent the quantitative result (B).

As far as the data of compound 8, it was omitted in the revised manuscript, because this compound should not be focused on in the present study.

4) About “In Figure 4, H2O2 decreased cell viability more than 50%, which is different than the data shown in Figure 2.”;

As mentioned in 1), neuro2a cells cultured in the presence of 10% FCS were exposed to 30 mM H2O2 for the experiment of Figure 2, while cells cultured in the presence of 2% FCS were exposed to 30 µM H2O2 for the experiment of Figure 4. We added the sentence in the revised manuscript (p.3, lines 141-145).

4) About “Why the authors have used higher doses of H2O2 when it already reached maximum cell viability inhibition.”;

   As pointed by reviewer, 30 µM H2O2 showed maximum ability on cell viability. We thus omitted the data with 60 µM H2O2 and 100 µM H2O2 in the revised manuscript.

5) About “In Figure 5 the western blot data is not the representative blot coming from a single gel but rather from more that one gel. Now a days very good pERK and pAKT antibodies are available to probe with for excellent results.”;

As pointed by reviewer, we had forgotten to show n number. We mentioned n number in the legend of Figure 5 (p.6, line 268-269). As for antibodies, we would like to offer our thanks for your assistance, but we have so far even longer purchased these antibodies from Cell Signaling Technology Inc. We believed that these antibodies are reliable.   

6) About “Finally, the authors didn’t discuss how the compound 13 is mediating its protective effect via PI3/ ERK kinases. Whether this compound 13 is binding to any G-protein coupled receptors, growth factor receptors or directly activating PI3/ ERK kinase pathways?”;

As pointed by reviewer, “other various signal pathways such as c-jun N-terminal kinase (JNK) have been reported to be involved in neurite outgrowth of neuro2a cells [19,21].” (p.6, lines 281-283). We thus also added the sentence “We will investigate whether not only PI3K/Akt-mediated signaling pathway but also other signaling pathway(s) underlie compound 13-induced neuronal differentiation or not.” (p.6, lines 282-286). 

Reviewer 2 Report

The article needs revision. First of all, in my opinion, the authors need to apply adequate statistical methods to assess the reliability of differences in results. If to assess the effect on the  cells viability of a number of different carbazole derivatives in one concentration, applying Student' t test is legitimate, then assessing the effect of different concentrations of compounds (leader and comparison) at different concentrations of toxicity inducer or assessing the influence of the leader compound on the phosphorylation of kinases depending on the incubation time requires other methods of statistical analysis - one- and two-way ANOVA.

In addition, in Figure 4, obviously, there is an error - it does not seem likely that there are significant differences from the control (30 μM H2O2) samples with H2O2 and 1 or 5 μM of compound 13.

In addition, the authors write in the conclusion: «To the best of our knowledge, this is the first study to show that a 363 carbazole derivative had neuroprotective effects». This is not true. A simple PubMed search shows that among the derivatives of carbazoles there are compounds with neuroprotective activity, as well as compounds with pro-neurogenic activity. It seems to me that adding this data to the introduction of the article could improve the article.

Author Response

The answer to Reviewer 2: 

According to the kind suggestion of reviewer 2, we collected our manuscript as follows.

1) About “the authors need to apply adequate statistical methods to assess the reliability of differences in results.”;

As pointed by reviewer, we applied only Student' t test as statistical methods in the previous manuscript. In the revised manuscript, we used Student’s t-test to analyze two groups, and a one-way ANOVA followed by the Dunnet’s multiple comparison test to analyse three or more groups. We explained in Materials and Methods 3.6. (p.5 line 363-365), figure legends (Figure 3, 4  and 5).    

2) About “in Figure 4, obviously, there is an error - it does not seem likely that there are significant differences from the control (30 μM H2O2) samples with H2O2 and 1 or 5 μM of compound 13.”;

   When we analyzed the differences of samples with H2O2 and 1 or 5 μM of compound 13 with a one-way ANOVA followed by the Dunnet’s multiple comparison test in the revised manuscript, the values were 0.0035 and 0.0033, respectively. As pointed by reviewer 1, 30 µM H2O2 showed maximum ability on cell viability. We thus omitted the data with 60 µM H2O2 and 100 µM H2O2 in the revised manuscript.

3) About “the conclusion: «To the best of our knowledge, this is the first study to show that a 363 carbazole derivative had neuroprotective effects».”;

   As pointed by reviewer, we searched PubMed and found some reports to show carbazole compounds with neuroprotective activity. We are sincerely sorry for our mistake. We omitted such sentence in Conclusion of the previous manuscript. As suggested by reviewer, we cited these reports (References 32-36) in Results and Discussion (p.7 lines 316-320).  

Round 2

Reviewer 1 Report

I understand the reason authors didn’t see the cell death since they have used 10%FCS in their cell culture and used the series of carbazole derivatives to examine the cell viability. In my opinion, the experiment 2 is not showing the protective effects of the carbazole derivatives rather they are not effective in inducing oxidative stress or cell death. Therefore, they can do two things;

1.   The authors are suggested to rewrite their statement that “The compounds 8, 13, 21 and 22 are not effective in inducing cell death in this experimental condition and gave similar cell viability in comparison to control and vitamin E treated cells”.

2.   The authors are suggested to remove the Figure 2 to make the things clear and reduce the discrepancies. They can use the same data to work on another manuscript showing the toxicity of the other compounds (ex – 12) in inducing cell death in cancer cell lines.

Author Response

The answer to Reviewer 1:   

According to the helpful suggestion of reviewer 1, we collected our manuscript as follows.

About “the neuroprotective effect of the compounds 13”;

As pointed out by reviewer, “the neuroprotective effect” was excessive expression. We thus changed the sentence to “had the protective effect on H2O2-induced cell viability reduction in this experimental condition and gave similar cell viability in comparison to that of control and 50 µM vitamin E (V.E)-treated cells” (p.2, lines 70-72) and “the protective effect on H2O2-induced cell viability reduction” (p.3, line 135).